# Current Development of Therapeutic Vaccines in Lung Cancer

**DOI:** 10.3390/vaccines13020185

**Published:** 2025-02-14

**Authors:** Jesus Salvador Flores Banda, Sanjana Gangane, Fatima Raza, Erminia Massarelli

**Affiliations:** Department of Medicine, University of Texas at Tyler School of Medicine, 11937 US Hwy 271, Tyler, TX 75799, USA; jesus.floresbanda@uttyler.edu (J.S.F.B.); sanjana.gangane@uttyler.edu (S.G.); fatima.raza@uttyler.edu (F.R.)

**Keywords:** lung cancer, small cell lung cancer, non-small cell lung cancer, therapeutic cancer vaccines, clinical trials

## Abstract

Cancer vaccines have a potential to change the current landscape of immunotherapy research and development. They target and neutralize specific tumor cells by utilizing the body’s own immune system which offers a promising modality in treating various cancers including lung cancer. Historically, prior vaccination approaches specifically towards lung cancer have posed several challenges but also potential with early phase I/II trials showing improved overall survival. With better understanding of the body’s immune system as well as advancements in vaccine development, the use of vaccines to target lung cancer cells in both non-small cell lung cancer (NSCLC) and small cell lung cancer (SCLC) has shown promise but also challenges in the setting of advanced stage cancers, tumor resistance mechanisms, immune evasion, and tumor heterogeneity. The proposed solution is to enroll patients in the early stages of the disease, rather than waiting until progression occurs. Additionally, future efforts will focus on the targeted identification of specific and novel tumor neo-antigens. This review offers discussion and analysis of both completed and ongoing trials utilizing different strategies for vaccine development in relation to treating lung cancer as well as current challenges faced.

## 1. Introduction

Lung cancer constitutes a significant global health challenge, ranking as the second most frequently diagnosed cancer and serving as the leading cause of cancer-related mortality in the United States [1]. While conventional treatment options often exhibit limited efficacy, cancer immunotherapy has emerged as a promising therapeutic avenue [2]. In this context, vaccines represent an essential strategy for activating the adaptive immune system against tumor cells [3]. Currently, there are four FDA-approved vaccines for cancer: sipuleucel-T for prostate cancer, bacillus Calmette-Guérin and nadofaragene firadonevec for bladder cancer, and T-VEC for melanoma, but no vaccines have been approved for lung cancer [4]. This article provides a comprehensive review of the latest advancements in therapeutic cancer vaccines specifically designed for non-small cell lung cancer (NSCLC) and small cell lung cancer (SCLC). By exploring recent clinical findings and ongoing trials, this article seeks to shed light on the potential of these vaccines to improve outcomes for patients diagnosed with lung cancer [5].

## 2. NSCLC Therapeutic Vaccines

NSCLC is categorized into three primary histological subtypes: lung adenocarcinoma, squamous cell carcinoma, and large cell carcinoma [6]. It is the most common form of lung cancer, but the complexity of this disease is further exacerbated by considerable genetic heterogeneity [7]. Unfortunately, NSCLC is associated with a challenging prognosis, evidenced by a five-year survival rate of only 15.9%, which has seen some improvements over recent decades due to approvals of immunotherapy and targeted therapies against specific genes including epidermal growth factor receptor (EGFR) and echinoderm microtubule-associated protein-like 4 (EML-4)-anaplastic lymphoma kinase (ALK), which have shown greater treatment response and increased survival in early stage and metastatic setting. Significant discoveries have led to targeted therapy being used in NSCLC with other specific gene alterations in the metastatic disease: KRASG12C, BRAFV600E, MET exon 14 skipping gene alterations, ROS1 fusions, RET fusions, and NTRK1-3 fusions [8]. For patients lacking actionable gene alterations, immune checkpoint inhibitors (ICIs) play a crucial role in treatment alone or in combination with platinum-based chemotherapy. Currently, surgery is potentially the most curative option for early-stage disease either with or without adjuvant, neoadjuvant, or perioperative chemotherapy in combination with immunotherapy or targeted therapy [9]. Additionally, tumor vaccines are emerging as a significant advancement in cancer therapies [10]. These interventions are part of a broader array of innovative immunotherapy strategies aimed at enhancing the immune system’s response to cancer cells.

### 2.1. Peptide/Protein Vaccines

A peptide vaccine represents a promising therapeutic strategy that employs synthesized protein fragments, referred to as peptides, to elicit targeted immune responses [11]. This approach involves the administration of tumor antigen-derived peptides in a personalized manner, effectively directing the immune system toward specific tumor antigens [12], as shown in Figure 1. One advantage of peptide and protein vaccines is that they can be administered intravenously, intradermally, or intramuscularly with some tolerable side effects, as stated in Table 1.

#### 2.1.1. Epidermal Growth Factor (EGF)

EGF is a signaling molecule that binds to and activates the EGFR on cell surfaces to promote proliferation and migration, but dysregulation of this process can contribute to many different cancers [13]. The vaccine utilizes a laboratory-engineered EGF protein to trigger an immune response, and this response results in the production of antibodies that target the body’s own EGF molecules, leading to reduced EGF levels in the bloodstream and diminished interaction with its receptor, EGFR [14]. The CIMAvax-EGF vaccine has received approval as a maintenance therapy for patients with stage IIIB/IV NSCLC who exhibit a partial response (PR) or stable disease (SD) following first-line chemotherapy in various Latin American countries. The treatment demonstrates significant benefits for patients exhibiting high serum EGF concentrations, specifically those exceeding 870 pg/mL [15]. Clinical findings indicate that this vaccine is associated with a median overall survival (OS) of 14.6 months (95% CI 10.6–18.8) and a progression-free survival (PFS) of 8.16 months (95% CI 4.9–11.3) [16]. The efficacy of this mechanism can be further improved when combined with ICIs, which facilitate the release of inhibitory signals on T cells, leading to more effective therapeutic outcomes [17]. The ongoing phase II clinical trial (NCT02955290) evaluates the efficacy of CIMAvax-EGF in combination with nivolumab in the American population. Preliminary results indicate OS of 11.9 months (90% CI 8.0–23.9), which appears to be a numerical improvement compared to historical cohorts receiving nivolumab therapy. Patients who are EGFR/ALK/KRAS wild-type experienced more favorable outcomes, as indicated by a median OS of 31.7 months (90% CI 5.9, NR) and a 3-year survival rate of 50% (90% CI 24, 71). In contrast, KRAS-mutated patients had a median OS of only 10.1 months (90% CI 6.5, 12.1) and a 3-year survival rate of 0% (90% CI 1, 41). Additionally, research shows that patients with PD-L1 expression levels of 1% or higher had better outcomes, with a 3-year OS rate of 38% (90% CI 12, 63) and a 3-year PFS rate of 38% (90% CI 12, 63). Conversely, patients without PD-L1 expression had lower rates, reporting a 3-year OS of 23% (90% CI 8, 44) and a 3-year PFS of just 8% (90% CI 1, 25) [18].

#### 2.1.2. Melanoma Antigen Family A, 3 (MAGE-A3)

MAGE-A3 is a tumor-specific shared antigen protein exclusively associated with neoplastic tissues [19]. It is not expressed in normal adult tissues, apart from testicular cells, which do not present the antigen to the immune system [20]. The expression of MAGE-A3 has been observed in various cancer types, including melanoma (74%), bladder cancer (35%), head and neck cancers (49%), and NSCLC (35–50%) [21]. The mechanism of action of the vaccine involves dendritic cells (DCs) capturing antigens and processing them into peptides that bind to HLA class I and II molecules. This interaction activates CD8+ and CD4+ lymphocytes, triggering an immune response against cancer cells [22]. The phase III clinical trial, referred to as the MAGRIT study (NCT00480025), has been terminated due to the Independent Data Monitoring Committee’s assessment that the study product did not demonstrate sufficient effectiveness [23]. However, preliminary findings from the early phase II clinical trial suggested that patients with MAGE-A3 positive resected NSCLC at stages IB, II, or IIIA experienced a 43% reduction in the relative risk of relapse compared to those receiving a placebo [24].

#### 2.1.3. New York Esophageal Squamous Cell Carcinoma 1 (NY-ESO-1)

NY-ESO-1 is a well-recognized cancer/testis antigen (CTA) known for its unique expression and immunological characteristics [25]. It has become a key target for cancer immunotherapy, especially for solid tumors like neuroblastoma (82%), synovial sarcoma (80%), melanoma (46%), cervical cancer (45%), and epithelial ovarian cancer (43%), with the vaccine primarily stimulating both cellular and antibody-based immune responses [26,27]. Two clinical trials assessing this vaccine, namely LV305 (NCT02122861) and G305 (NCT02015416), intended to recruit patients with solid tumors, including NSCLC. However, only one patient with NSCLC was enrolled in the LV305 trial and discontinued treatment due to disease progression [28]. In the latter trial G305, patients with NSCLC have still to be enrolled [29].

#### 2.1.4. Mucin 1 (MUC1)

Mucin 1 (MUC1) is a transmembrane protein characterized by extensive glycosylation, which plays a crucial role in various physiological functions, including protection and lubrication within the body [30]. In cancer cells, however, MUC1 exhibits abnormal glycosylation patterns and increased expression levels, thereby altering its role from a protective molecule to one that facilitates tumor progression [31]. A promising therapeutic vaccine, L-BLP25, commercially also known as tecemotide, has been developed as an immunotherapy for NSCLC, specifically targeting MUC1 to enhance treatment efficacy [32]. The phase I clinical trial (NCT01731587), which focused on patients with unresectable stage III NSCLC demonstrating SD following primary chemoradiotherapy, has been withdrawn because the objectives of the trial were found to not be suitable for the development of the vaccine [33]. In contrast, the phase IIb clinical trial (NCT00828009) assessed the efficacy of tecemotide in combination with bevacizumab as maintenance therapy for patients with unresectable stage IIIA and IIIB non-squamous NSCLC following definitive chemoradiation. This trial reported OS of 42.7 months (95% CI 21.7–63.3) and PFS of 14.9 months (95% CI 11.0–20.9) [34]. Conversely, the findings of the phase III START trial (NCT00409188), which evaluated the vaccine alone, indicated no significant difference in OS between the vaccinated patients and the control group, with median survival durations of 25.6 months (95% CI 22.5–29.2) and 22.3 months (95% 19.6–25.5), respectively [35]. Subsequent trials, including the START2 trial (NCT02049151) which aimed to evaluate tecemotide in patients with unresectable stage III NSCLC as well as the INSPIRE trial (NCT01015443), which focused primarily on the Asian population, both were discontinued prematurely [36,37]. This decision was made by the sponsor due to the studies not meeting the primary endpoint OS as well secondary endpoints which were PFS, time to progression, and time to treatment failure [38].

### 2.2. DNA Vaccines

A DNA vaccine is an innovative approach to immunization that utilizes genetic material in the form of engineered plasmids (Figure 2). These plasmids carry instructions for producing specific target proteins that stimulate immune responses [39]. One of the major advantages of DNA vaccines is their ability to engage multiple components of the immune system simultaneously. They present their encoded antigens through both MHC class I and II pathways, leading to the activation of CD8 cytotoxic T cells and CD4 helper T cells, while also stimulating antibody-based (humoral) immune responses [40]. However, no current clinical trials are recruiting patients with NSCLC [41].

### 2.3. Vector Vaccines

A vector vaccine represents a form of genetic vaccination that employs a modified virus as a delivery mechanism to introduce genetic material encoding disease-specific antigens into human cells [42] (Figure 3). Utilizing the inherent capability of viruses to transport genetic material into cells, these vaccines facilitate the production of specific antigens that stimulate immune responses [43]. The main advantage of vector vaccines is correlated to their ability to elicit both cellular and humoral immune responses, to overcome the need for additional adjuvants [44]. Another advantage of vector vaccines is that they can be administered either intravenously or intramuscularly, and the reported side effects are manageable, as stated in Table 2. Several clinical trials are ongoing to evaluate the efficacy of vector vaccines (Table 2). A phase II clinical trial (NCT03353675) enrolled patients with advanced, untreated NSCLC with less than 50% PD-L1 expression after first-line chemotherapy; participants received the vector vaccine MVA-MUC1-IL2, also named as TG410, which includes a modified virus carrying the MUC1 protein and interleukin-2, alongside nivolumab, but the results have not been reported yet [45]. A phase II European clinical trial assessed the effects of concurrent chemotherapy and TG410 in two arms. Arm 1 received chemotherapy plus the vaccine, while arm 2 started with the vaccine alone and later added chemotherapy upon disease progression. Arm 1 showed a PR in 13 out of 37 patients with stage III/IV NSCLC, based on Response Evaluation Criteria in Solid Tumors (RECIST) criteria, while arm 2 did not meet criteria for proceeding to the second state [46]. A new vector vaccine composed with semi-allogeneic fibroblasts transfected with autologous tumor-derived DNA was about to be evaluated in the clinical trial (NCT00793208), but the study was terminated due to low enrollment [47]. The MG1-MAGEA3 with Ad-MAGEA3 vector vaccine combines an engineered oncolytic Maraba virus that expresses MAGE-A3 with transgenic MAGE3 adenovirus vaccines [48,49]. The phase II clinical trial (NCT02879760) assessed the efficacy of a vaccine combined with pembrolizumab in patients with squamous and non-squamous NSCLC who had completed at least one cycle of platinum-based chemotherapy and/or at least one treatment with PD-1 or PDL-1 antibody-targeted therapy. Results from the trial have not yet been reported [50]. An active phase I clinical trial (NCT02285816) is assessing MG1-MAGEA3 with and without Ad-MAGEA3 in patients with incurable, advanced, or metastatic MAGE-A3-expressing solid tumors, including NSCLC (adenocarcinoma and squamous cell carcinoma), although results have not yet been reported [51]. Pitfalls of viral-based cancer vaccines are related to risk of side effects from the viral vector, such as inflammation and allergic reactions. In addition, effectiveness of these vaccines can be reduced by pre-existent immunity to the virus used to construct the vaccine.

### 2.4. Dendritic Cell (DC) Vaccines

A DC vaccine is an advanced immunotherapy that utilizes the unique ability of DCs to activate CD4+ and CD8+ T cells, enhancing the immune response to tumor-specific antigens [52] (Figure 4). This approach addresses the challenge of inadequate antigen presentation in cancer patients [53]. The process typically involves isolating DCs from the patient’s monocytes or CD34+ cells, loading them with tumor antigens, and activating them with toll-like receptor ligands or cytokines, then the activated DCs are then reintroduced into the patient to stimulate a robust immune response against the cancer [54]. There are several clinical trials assessing the efficacy of DC vaccines (Table 3). A phase I clinical trial (NCT00601094) is aiming to assess the safety, toxicity, and maximum tolerated dose of autologous dendritic cell-adenovirus CCL21, known as AdCCL21-DC, in patients with stage III/IV and recurrent NSCLC [55]. Results have not yet been reported. Another phase I clinical trial (NCT01574222) found that the vaccine was safe, with no evidence of free adenovirus in the peripheral blood following vaccination, and no significant changes in anti-adenoviral antibody levels were detected. A systemic immune response was found in 6 out of 16 patients and was directed against tumor-associated antigens (TAAs) as demonstrated through ELISPOT assays. Additionally, there was an average increase of 3.4 times in CD8 T-cell infiltration per mm^2^, with 25% of the patients (4 out of 16) showing a clinical response of SD by day 56 [56]. An ongoing clinical trial (NCT03546361) is investigating the efficacy of pembrolizumab in combination with AdCCL21-DC in stage IV NSCLC patients who are either EGFR/ALK wild-type after progression on a PD-L1 inhibitor or EGFR/ALK mutant after progression on TKI therapy. Preliminary studies indicate the infiltration of autologous DCs and T cells into the tumor microenvironment (TME), generating systemic tumor-specific immune responses against multiple tumor antigens [57]. Disadvantages of these vaccines are related to their costly and complex production, therefore jeopardizing their wide range utilization. Another limitation is represented by the need for intratumoral administration (i.e., via bronchoscopy) that can limit their application to hospital settings rather than community settings. Side effect profile appears tolerable, as reported in Table 3. The limited efficacy observed in DC vaccines can be attributed to suboptimal protocols that fail to generate an optimal T-cell response. One factor contributing to this limitation is the use of granulocyte-macrophage colony-stimulating factor (GM-CSF) for maturing peripheral blood mononuclear cells, which results in the production of monocyte-derived DCs that have a limited ability to migrate to lymph nodes [37,38]. The migration of DCs to lymph nodes is crucial for T-cell antigen interaction, and the use of monocyte-derived DC vaccines hampers this essential step [39,40,41].

Furthermore, it has been highlighted by Roy et al. (2020) [58] that the production of DC vaccines can be costly and complex, further posing challenges to their widespread adoption [58]. These factors have contributed to the need for refinement in the manufacturing and administration of DC vaccines to enhance their effectiveness.

### 2.5. Allogenic Vaccines

An allogeneic vaccine is a type of cancer vaccine made from tumor cells that are harvested from donors rather than the patient themselves [59] (Figure 5). These vaccines work by stimulating immune responses, specifically through T cells that recognize certain tumor antigens [60]. To ensure safety, the tumor cells are usually irradiated or destroyed before being reinjected into patients, so this process helps to minimize the risk of tumor cell implants or metastasis [61]. Numerous clinical trials are currently investigating various allogenic vaccines for oncological applications (Table 4). A phase III clinical trial (NCT00676507), the STOP trial, compared the effectiveness of belagenpumatucel-L with a placebo in patients diagnosed with stage III/IV NSCLC. Unfortunately, this trial did not meet its primary objective, as there was no statistically significant difference in median OS between the two groups. The vaccinated group had a median OS of 20.3 months, while the placebo group had a median OS of 17.8 months (HR 0.94, *p* = 0.594). Additionally, the trial did not meet its secondary endpoint for PFS, with the vaccinated group experiencing a median PFS of 4.3 months versus 4.0 months for the placebo group (HR 0.99, *p* = 0.947) [62]. A phase II clinical trial (NCT00601796) evaluated GM.CD40L in combination with cyclophosphamide and tretinoin in patients with stage IIIB/IV lung adenocarcinoma; however, it failed to meet its primary endpoint of inducing radiologic tumor regression even in patients with an immune response to vaccination [63]. Another phase Ib/II2 clinical trial (NCT02460367) assessed tergenpumatucel-L in previously treated NSCLC patients but was terminated prematurely due to insufficient enrollment [64]. Additionally, a phase II clinical trial (NCT01774578) is currently underway to evaluate tergenpumatucel-L in comparison to docetaxel for patients with advanced NSCLC, although the findings have yet to be disclosed [65]. The DURGA trial (NCT02439450), a phase Ib/2 study, examined the safety profile of viagenpumatucel-L in combination with nivolumab. Preliminary results suggest that this combination is safe for patient use [66].

The main advantage found in allogenic vaccine consists in the enhanced immune response that mimics graft versus tumor responses observed in patients who receive mismatched minor histocompatibility antigen bone marrow transplants [67]. Other advantages are availability, low production cost, and the lack of invasive procedures as they are administered intradermally with tolerable side effects as reported in Table 4. Disadvantages are represented by the fact that irradiated cells may retain the ability to secrete immunosuppressive factors like the original tumor cells [68,69].

### 2.6. Messenger RNA (mRNA) Vaccines

mRNA vaccines represent a significant advancement in biotechnology by delivering genetic information that encodes target antigens in the form of mRNA, which is subsequently translated within the cytoplasm of cells [70]. Upon entry into the cells, ribosomes translate the mRNA into proteins, which are then presented on the surface of the host cells [71] (Figure 6). These vaccines gained widespread recognition during the COVID-19 pandemic, as they became the first approved vaccines with a preventive efficacy of 95% [72]. The expedited development of mRNA vaccines for COVID-19 has highlighted the remarkable potential of mRNA technology, which is currently being explored for use in cancer treatments, leading to notable clinical advances in ongoing trials [73] (Table 5). These vaccines also offer significant advantages by addressing key limitations of traditional vaccines. They provide information on tumor-specific antigens, allowing for precise identification and targeting cancer cells while also fostering robust and lasting immune responses against tumors [74]. The phase I clinical trial (NCT05142189), known as LuCa-MERIT-1, currently recruiting, is testing the mRNA vaccine BNT116 in patients with advanced NSCLC who have a PD-L1 expression of ≥50% after receiving anti-PD-L1 therapy. The vaccine is composed of six mRNAs (MAGE A3, CLDN6, KK-LC-1, PRAME, MAGE A4, MAGE C1), each of which encodes tumor-associated antigens (TAAs) commonly found in NSCLC. Preliminary results indicated that 35% of patients had PR and 50% had SD, according to the RECIST v1.1 criteria [75]. In addition, there are two phase III clinical trials, NCT06623422 and NCT06077760, that are actively recruiting patients with resected stage II, IIIA, and IIIB NSCLC. These trials aim to evaluate the mRNA vaccine V940 in combination with pembrolizumab in the adjuvant space, compared to pembrolizumab plus placebo [76,77]. Results from these trials have not yet been reported. Advantages of mRNA vaccines are high potency, safe administration, rapid development potential, and cost-effective manufacturing.

### 2.7. Neoantigens Vaccines

Neoantigens, short peptide sequences expressed only by tumor tissues, are unique to each patient and tumor. This option provides a solution to challenges posed by resistance to current prevailing standard of care treatments such as ICIs, commonly utilized in the treatment of NSCLC. While the vaccines discussed thus far in this article are personalized in that they target TAAs, they have been conducted on a relatively small number of patients, and larger scale prospective studies are required. There are several studies underway to facilitate development of neoantigen vaccines. One such example is the phase I/II clinical trial NCT03953235, the purpose of which is to evaluate the safety, dose, immunogenicity and early clinical activity of GRT-C903 and GRT-R904, a personalized neoantigen cancer vaccine, in combination with nivolumab and ipilimumab, in patients with metastatic non-small cell lung cancer, microsatellite stable colorectal cancer, gastroesophageal adenocarcinoma, and metastatic urothelial cancer. This study was completed in 2024, and its results are still pending [78]. Another example is NCT03908671, testing a personalized RNA vaccine encoding neoantigens in the treatment of NSCLC [79].

## 3. SCLC Therapeutic Vaccines

SCLC comprises 15% of all lung cancer diagnoses and is recognized as one of the most aggressive forms of lung cancer [80]. It accounts for about 15% of all lung cancers worldwide and is notable for its propensity for rapid proliferation and early metastasis. Despite the initial positive response to treatments like chemotherapy and radiation, SCLC poses a dire threat, as the majority of patients experience recurrence often accompanied by metastasis to distant sites, leading to a staggering survival rate of under 5% over five years, while those not receiving any active treatment usually survive just 2 to 4 months [81]. The treatment landscape for SCLC has seen limited advancements, with its first-line therapy remaining unchanged for over three decades; however, recent developments offer a renewed sense of optimism [82]. Recently, there has been an increase in understanding of the molecular alterations causing SCLC, leading to the integration of immunotherapy in standard treatment regimens. The integration of ICIs such as atezolizumab and durvalumab targeting PD-L1 into first-line treatment regimens has been demonstrated to improve both OS and PFS [83]. Recent developments have led to the approval of the bispecific T-cell engager (BiTE) tarlatamab after failure of first-line treatment [84]. PARP inhibitors and antibody drug conjugates targeting DLL3, a NOTCH ligand, represent emerging therapies showing promising results in ongoing trials [85]. Notably, the exploration of vaccine therapies presents a biologically driven approach aimed at addressing residual cancer cells, underscoring potential progress in the management of this challenging disease [86]. Table 6 lists completed and current clinical trials of cancer vaccines in SCLC.

### 3.1. Combined Vaccine with the Bacillus Calmette-Guérin

Bacillus Calmette-Guerin (BCG) is mainly used to treat non-muscle-invasive bladder cancer [87]. It works by activating natural killer (NK) cells and CD8+ T cells, which helps boost the immune response against cancer [88]. This live vaccine has also been studied for use in other types of cancer. For instance, the Silva trial, a completed phase III clinical study (NCT00037713, NCT00003279, NCT00006352), assessed the effectiveness of BEC-2, an anti-idiotypic antibody targeting GD3, combined with BCG in 515 patients with limited disease SCLC [89,90,91]. The study was performed between March 1998 and October 2002 [92,93]. However, the results showed no significant improvements in survival rates (16.4 and 14.3 months *p* = 0.28), PFS, or quality of life for the vaccination group [94]. One advantage of this vaccine is that it is administered intradermally and has tolerable side effects.

### 3.2. Tumor Glycans Vaccines

Tumor-associated carbohydrate antigens (TACAs), such as O-glycans, gangliosides, globo-series glycans, Lewis antigens, and polysialic acid, are produced through abnormal glycosylation processes in tumor cells and play a crucial role in metastasis and signal transduction [95]. Vaccines that incorporate TACA antigens are absorbed by antigen-presenting cells via endocytosis. Once inside the cells, these antigens are processed and presented through the MHC class I and II pathways, which stimulates an immune response against cancer [96]. When combined with other anti-tumor strategies, these vaccines represent a promising approach to cancer treatment [97]. The pentavalent KLH conjugate vaccine contains five antigens: GD2L, GD3L, Globo H, fucosyl GM1, and N-propionylated polysialic acid, all linked to an immunostimulant called keyhole limpet hemocyanin (KLH) [98]. The completed phase 1 clinical trial, NCT01349647, tested the pentavalent KLH conjugate vaccine in patients with SCLC, but results have not been posted yet [99].

### 3.3. Peptide/Protein with Dendritic Cell (DC) Vaccine

Numerous cancer-specific antigen vaccines have been developed for the treatment of SCLC, either alone or in combination with DCs. Creating a DC vaccine involves several steps. First, DCs are separated from the patient’s blood. They are then activated and loaded with antigens outside the body (ex vivo) before being reinfused into the patient to stimulate an immune response against cancer [100]. One of the antigens explored in tumor vaccine development is P53, a crucial regulator of various cellular processes, including cell growth, immune responses, and multiple signaling pathways [101]. Its inactivation is almost universally observed in human cancer cells, significantly impacting advancements in cancer therapeutics [102]. P53-based peptide vaccines are designed to trigger an anti-tumor immune response by targeting the p53 protein [103]. Another antigen explored in this field is the RAS protein, which plays an important role in signal transduction within cellular membranes, but dysregulation of this protein can lead to cancer [104]. RAS peptide cancer vaccines aim to enhance the activation of a RAS peptide-specific anti-tumoral T-cell cytotoxic immune response, which may help inhibit tumor cell growth and promote tumor cell death [105]. A completed phase II clinical trial, referred to as NCT00019084, evaluated the efficacy of P53 or RAS peptide vaccines, either independently or in combination with the patient’s own peptide-activated lymphocytes and subcutaneous administration of interleukin-2, a cytokine that enhances the immune response in cancer therapy [106,107]. One advantage of these vaccines is that they are administered intravenously. However, the results of this trial have yet to be disclosed. The Ad.p53-DC vaccine is a type of cancer immunotherapy that involves using a patient’s DCs that have been modified with a recombinant adenovirus carrying the p53 peptide, which may enhance immune response [108]. Numerous clinical trials are currently underway to assess the Ad.p53-DC vaccine. The phase II clinical trial NCT03406715 is investigating the Ad/p53-DC vaccine in conjunction with immunotherapeutic agents nivolumab and ipilimumab [109]. Two clinical trials, NCT00049218 (phase I/II) and NCT00617409 (phase II), are focused on evaluating the effectiveness of chemotherapy followed by the administration of the Ad.p53-DC vaccine in patients with extensive-stage SCLC [110,111]. The phase II clinical trial also explores the combination of chemotherapy with or without all-trans retinoic acid, a compound that the human body produces from vitamin A and that aids in cell growth and development [112]. One advantage of this vaccine is that it is administered intradermally. The phase I/II clinical trial NCT00049218, also known as INGN-225, assessed the safety and tolerability of the Ad.p53-DC vaccine in 54 patients with SCLC. Clinical responses correlated with positive immune responses in 11 out of 14 (79%). In contrast, only 5 out of 15 (33%) patients with a negative immune response met the RECIST criteria for clinical response (*p* = 0.014) [113]. In general, advantages of peptide vaccines are ease of synthesis, low production costs, low carcinogenic potential, high chemical stability, and tolerable side effects (Table 6). The disadvantages are low immunogenicity, requiring combination therapies.

## 4. Conclusions

This review provides an analysis of various strategies for vaccine development, summarizing results of both completed and ongoing clinical trials. These efforts are crucial in enhancing treatment options for both NSCLC and SCLC patients. Despite the progress made in research and development, many trials have encountered significant challenges in meeting their primary endpoints, reflecting the inherent difficulties in effectively targeting these aggressive forms of cancer. Given that lung cancer remains the leading cause of cancer-related mortality among both men and women, with a concerning five-year survival rate of only 28.4% following diagnosis, there is a pressing need for innovative therapeutic strategies to enhance vaccine outcomes [114].

Currently, there are several promising vaccine clinical trials in progress aimed at improving the landscape of lung cancer treatment. However, it is noteworthy that, to date, only four therapeutic cancer vaccines have received FDA approval. Disturbingly, none of these vaccines are specifically designated for lung cancer treatment. Among the few approved options, the peptide and protein vaccines stand out, specifically CIMAvax-EGF vaccine in combination with or without nivolumab. The CIMAvax vaccine has been approved in Latin America, where it has demonstrated a median OS of 14.6 months (95% CI 10.6–18.8) and a PFS of 8.16 months (95% CI 4.9–11.3) for patients with NSCLC as a maintenance therapy [16]. The CIMAvax vaccine combined with nivolumab has shown an OS of 11.9 months (90% CI 8.0–23.9) in patients with late-stage NSCLC requiring systemic therapy [18]. The importance of this endeavor is immense, as progress in vaccine development could significantly enhance survival rates and the quality of life for individuals battling lung cancer.

## 5. Expert Opinion

A significant number of clinical trials conducted thus far in lung cancer have primarily engaged patients who have not responded to conventional first- and second-line therapies, as well as ICIs and tyrosine kinase inhibitors (TKIs). It is important to recognize that many patients are diagnosed at advanced stages of their diseases, including late-stage, unresectable, incurable, or metastatic conditions. This has a considerable impact on their prognosis and makes treatment options more complex. This group often exhibits signs of immunosuppression as a consequence of prior chemotherapy treatments and bulky disease burden correlated with an immunosuppressed TME, which can substantially impair their response to vaccine therapies. Therefore, a more promising strategy might involve the enrollment of patients at earlier stages of the disease or during the phase of early detection. This approach enables more timely and effective treatment through the utilization of immunotherapy options, such as combined ICIs and tumor vaccines, in conjunction with established chemotherapy regimens.

Another significant challenge that vaccine implementation faces is the variability of molecular gene alterations associated with tumorigenesis. Notably, the clinical trial NCT02955290 has demonstrated that patients with EGFR, ALK, or KRAS wild-type profiles achieve significantly better outcomes when treated with CIMAvax-EGF in combination with nivolumab, resulting in a median OS of 31.7 months (90% CI 5.9, NR) and a 3-year OS rate of 50% (90% CI 24, 71), whereas patients with KRAS mutations have a median OS of only 10.1 months (90% CI 6.5, 12.1) and a 3-year OS rate of 0% (90% CI 1, 41) [18]. This difference in outcomes may be due to the fact that KRAS-mutated lung cancers often develop multiple resistance mechanisms, which can alter the effectiveness of immunotherapy treatments, including tumor vaccines. The same study suggests that patients with PD-L1 expression levels of ≥1% may have significantly greater benefit from a combination of ICIs and tumor vaccination compared to their counterparts with PD-L1 ≤1% [18]. Therefore, it might be advisable to focus the development of optimal tumor vaccines on patients exhibiting PD-L1 expression levels of 1% or higher without driver gene alterations.

Additionally, tumor vaccines are instrumental in enhancing T-cell infiltration in tumors and antigen presentation. Therefore, appropriate sequencing of delivery of vaccines in respect to combined treatments such as ICIs may be a key factor in eliciting the strongest immune response as demonstrated in studies by Sousa et al. [115].

Investigating the immune response variability in NSCLC is essential for identifying significant biomarkers. A notable example is the CIMAvax-EGF vaccine, which has demonstrated improved outcomes for patients with elevated serum EGF levels [15]. Therefore, it is imperative to identify more precise markers that can predict which patients are likely to benefit from this therapy, particularly after the vaccine has established its efficacy. In fact, monitoring immune response at defined time points after vaccination is essential to decide if patients are benefiting from this approach preventing rapid progression of disease.

Novel approaches in developing vaccine delivery systems to overcome tumor heterogeneity involve the incorporation of artificial intelligence (AI), nanoparticle technology, and the use of embryonic stem cells (ESCs). AI can aid in extracting large datasets and training models to predict patient-specific cancer antigens as well as assessment of vaccine immunogenicity. This information can help guide the design and formulation of vaccines, with several models demonstrating promise in supporting traditional methods and holds potential for creating more personalized and targeted therapies [116].

The use of nanovaccines that employ nanoparticles, which range from 50 to 200 nm in diameter and serve as carriers and/or adjuvants, has been demonstrated to improve delivery effectiveness in vitro and in vivo [117]. These nanovaccines can boost the activation of CD8+ T cells by at least 55 times in comparison to soluble antigens, which can overcome immunosuppressive TME signals. They can also specifically target tumor sites and aid in delivery to the lymphatic system. Additionally, they can be tailored according to the unique characteristics of a patient’s tumor, providing potential for targeting heterogeneity in tumor metastasis [117]. Recent studies showed that irradiated, induced pluripotent stem cells (iPSCs) function as a prophylactic vaccine against transplanted tumors by eliciting anti-tumor immune responses [118,119]. Shared expression of antigens between ESCs and lung cancer cells might induce immunity against pulmonary malignancy [120].

Undifferentiated ESC vaccines appear to be more effective in preventing lung tumor development in a Lewis lung mouse model as they overexpress certain antigens, including carcinoembryonic antigen (CEA), prostate-specific antigen (PSA), and CTA that can enhance immunogenicity [121].

In conclusion, cancer vaccines represent useful additions to standard treatments such ICIs, chemotherapy, and radiation therapy to improve outcomes especially in early stage lung cancer in light of availability of innovative approaches such as AI, nanovaccines, and ESCs.

## Figures and Tables

**Figure 1 vaccines-13-00185-f001:**
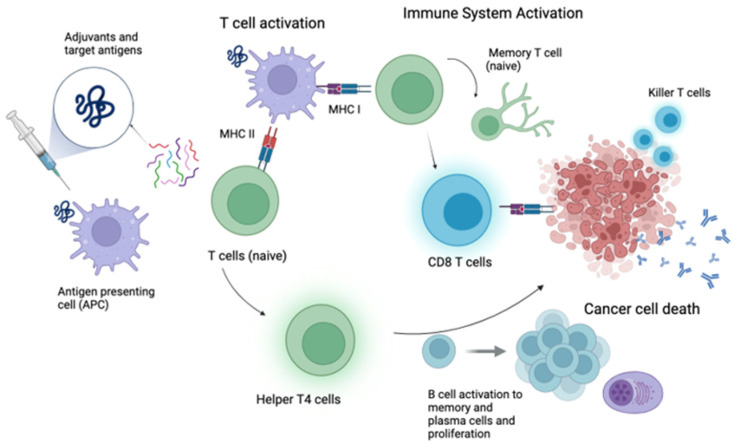
Mechanism of action of tumor peptide vaccines. MHC I: Major histocompatibility complex I, MHC II: Major histocompatibility complex II. Created in BioRender. Gangane, S. (2025) https://BioRender.com/h94t084 (accessed on 30 January 2025).

**Figure 2 vaccines-13-00185-f002:**
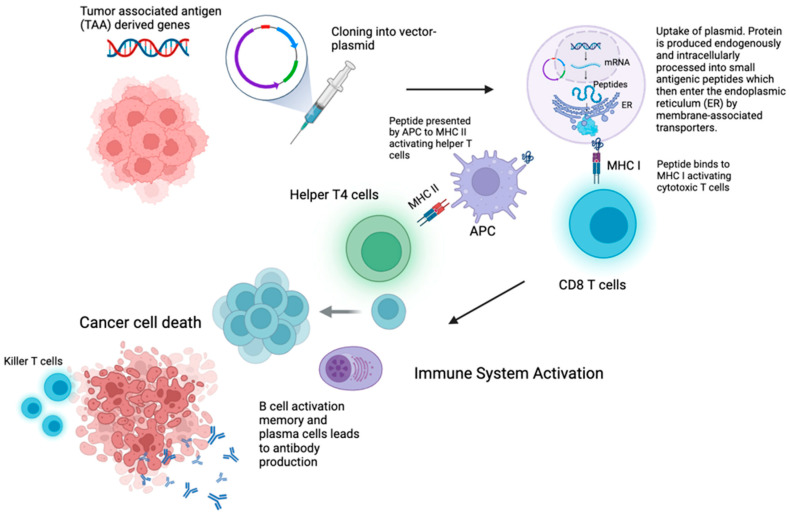
Mechanism of action of tumor DNA vaccines. MHC I, MHC II, APC: Antigen-presenting cell. Created in BioRender. Gangane, S. (2025) https://BioRender.com/y82k991 (accessed on 30 January 2025).

**Figure 3 vaccines-13-00185-f003:**
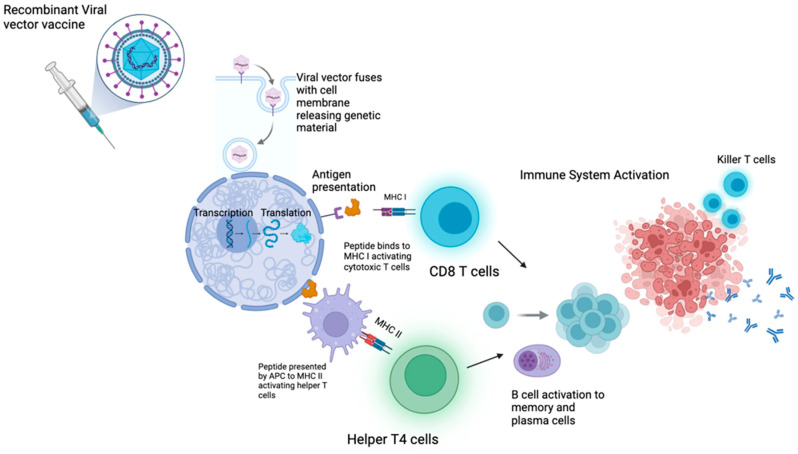
Mechanism of action of vector vaccines. MHC I, MHC II, APC. Created in BioRender. Gangane, S. (2025) https://BioRender.com/e73e024 (accessed on 30 January 2025).

**Figure 4 vaccines-13-00185-f004:**
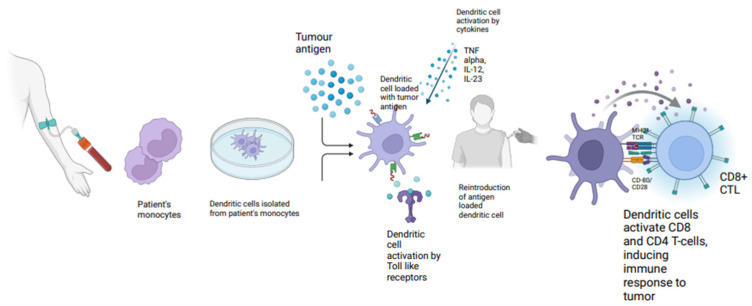
Mechanism of action of dendritic cell vaccines. MHC I, MHC II, TNF-α: Tumor necrosis factor alpha, IL-12: Interleukin 12, IL-23: Interleukin 23, TCR: T-cell receptor, CTL: cytotoxic T-lymphocyte. Citation: Created in BioRender. Raza, F. (2025) https://BioRender.com/c23d034 (accessed on 31 January 2025).

**Figure 5 vaccines-13-00185-f005:**
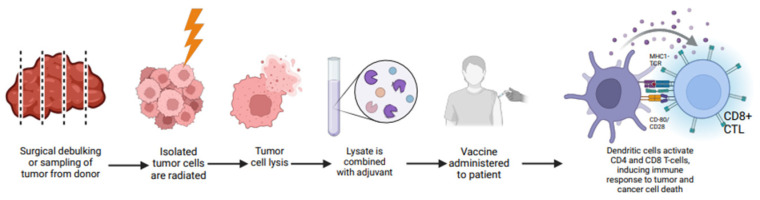
Mechanism of action of allogenic vaccines. MHC I, TCR, CTL. Citation: Created in BioRender. Raza, F. (2025) https://BioRender.com/c23d034 (accessed on 31 January 2025).

**Figure 6 vaccines-13-00185-f006:**
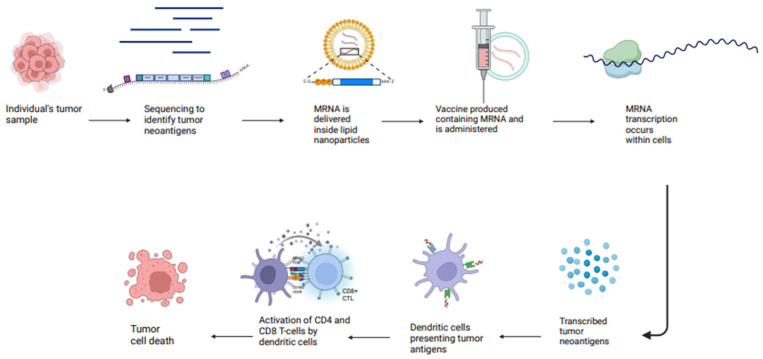
Mechanism of action mRNA vaccines. Citation: Created in BioRender. Raza, F. (2025) https://BioRender.com/c23d034 (accessed on 30 January 2025).

**Table 1 vaccines-13-00185-t001:** Previous and ongoing clinical trials with peptide vaccines.

Name (Clinical Trial)	Phase	Stage	Side Effects	Results
CIMAvax-EGF	Longitudinal study conducted in a real-world setting	IIIB/IV NSCLC with partial response or stable disease after first-line chemotherapy	Injection site pain, local erythema, nausea, dizziness, and fever	Primary endpoints - OS 14.6 months (95% CI 4.9–11.3) - PFS 8.16 months (95% CI 4.9–11.3)
CIMAvax-EGF in combination with nivolumab (NCT02955290)	II	Late-stage disease requiring systemic therapy	Information not published	Primary endpoint - OS 11.9 months (90% CI 8.0–23.9)
MAGE-A3 (NCT00480025-MAGRIT study)	III	MAGE-A3 positive resected IB, II, or IIIA NSCLC	Information not published	Study was terminated
LV305 (NCT02122861)	I	Locally advanced, relapsed, or metastatic cancers expressing NY-ESO-1 (≥5% of expression)	Fatigue, injection site reactions, and myalgia	Only one patient with NSCLC enrolled.
Tecemotide (NCT01731587)	Ib	Unresectable stage III NSCLC with stable disease after primary chemo-radiotherapy	Clinical trial withdrawn	Clinical trial withdrawn
Tecemotide in addition to bevacizumab as a maintenance therapy (NCT00828009)	IIB	Unresectable Stage IIIA and IIIB Non-Squamous NSCLC after definitive chemoradiation	Anemia, heart failure, pericardial effusion, esophageal ulcer, hemorrhoidal hemorrhage, sepsis, UTI, transaminitis, lymphopenia, weight loss, hypertension, and thromboembolic events	Primary endpoint - Safety and tolerability Outcomes - OS 42.7 months (95% CI 21.7–63.3) - PFS 14.9 months (95% CI 11.0–20.9)
Tecemotide (NCT00409188—START trial)	III	Unresectable stage III NSCLC who has completed chemoradiation	Dyspnea, metastases to central nervous system, and pneumonia	Primary endpoint - No change in median OS 25.6 months (95% CI 22.5–29.2) in the tecemotide group versus 22.3 months (95% 19.6–25.5) in the placebo group
Tecemotide (NCT02049151—START2 trial)	III	Unresectable stage III NSCLC receiving concurrent chemoradiation	Information not published	Study was terminated
Tecemotide (NCT01015443—INSPIRE trial)	III	Unresectable stage III NSCLC in the Asian population	Information not published	Study was terminated

OS: Overall survival, PFS: Progression-free survival.

**Table 2 vaccines-13-00185-t002:** Vector vaccine clinical trials.

Name (Clinical Trial)	Phase	Stage	Side Effects	Results
First-line chemotherapy combined with TG4010 and nivolumab (NCT03353675)	II	Advanced stage and immunotherapy-naive non-squamous NSCLS and with <50% of tumor cells expressing PD-L1	Information not published	Pending
TG4010 in association with chemotherapy	II	Stage IIIB and IV NSCLC	Fatigue, injection site reactions, injection site pain, pyrexia, and fever	Primary endpoint - Arm 1 PR was observed in 13/37 patients by RECIST criteria. - Arm 2 did not meet criteria for PR
New vaccine semi-allogeneic fibroblasts transfected with autologous tumor-derived DNA (NCT00793208)	I	NSCLC	Information not published	Study closed due to low enrollment.
MG1-MAGEA3 with Ad-MAGEA3 Vaccine in combination with pembrolizumab (NCT02879760)	I/II	NSCLC patients who have completed a first standard therapy with at least 1 cycle of chemotherapy and/or at least one treatment of PD-1 or PDL-1 antibody targeted therapy.	Information not published	Pending
MG1 Maraba/MAGE-A3, with and without adenovirus vaccine with transgenic MAGE-A3 insertion (NCT02285816)	I	Incurable advanced/metastatic MAGE-A3-expressing solid tumors	Information not published	Pending

PR: Partial response.

**Table 3 vaccines-13-00185-t003:** Dendritic cell vaccine clinical trials.

Name (Clinical Trial)	Phase	Stage	Side Effects	Results
AdCCL21-DC (NCT00601094)	I	Stage IIIB, stage IV, or recurrent NSCLC	Information not published	Pending
AdCCL21-DC (NCT01574222)	I	Stage IIIB, stage IV NSCLC	Flu-like symptoms, nausea, and fatigue	Primary endpoint - Safety - Manageable adverse events - Systemic immune response in 6/16 patients
AdCCL21-DC in combination with pembrolizumab (NCT03546361)	I	Stage IV NSCLC patients who are either EGFR/ALK wild-type after progression on a PD-L1 inhibitor or EGFR/ALK mutant after progression on TKI therapy	Information not published	Pending

**Table 4 vaccines-13-00185-t004:** Allogenic cell vaccine clinical trials.

Name (Clinical Trial)	Phase	Stage	Side Effects	Results
Belagenpumatucel-L (NCT00676507—STOP trial)	III	III/IV NSCLC patients who had no disease progression after first-line chemotherapy	Injection site reactions, cough, and fatigue	The study failed primary endpoint in median OS, 20.3 (95% CI, 16.8–23.7) for vaccinated group and 17.8 months (95% CI, 13.7–22.0) for placebo group.
GM.CD40L bystander cells admixed with an equivalent number of the 2 allogeneic tumor cell lines, NCI-H1944, NCI-H2122 in combination with cyclophosphamide and tretinoin (NCT00601796)	II	IIIB and IV lung adenocarcinoma	Headache, injection site reaction, and fatigue.	The study failed primary endpoint of inducing radiologic tumor regression even in patients with an immune response to vaccination.
Tergenpumatucel-L (NCT02460367)	Ib/2	Previously treated NSCLC	Information not published	Study was terminated
Tergenpumatucel-L (NCT01774578)	2B	Advanced NSCLC previously undergone surgery, radiotherapy, and ≤2 prior chemotherapy regimens for this diagnosis	Information not published	Pending
Viagenpumatucel-L in combination with nivolumab (NCT02439450—DURGA trial)	Ib/2	Advanced NSCLC	Fatigue, cough, and diarrhea	Primary endpoint - Combination is safe

**Table 5 vaccines-13-00185-t005:** mRNA vaccine clinical trials.

Name (Clinical Trial)	Phase	Stage	Side Effects	Results
BNT116 in addition to docetaxel (NCT05142189—LuCa-MERIT-1)	I	Advanced NSCLC, PD-L1 expression ≥ 50%, disease progression on or after anti-PD-L1 therapy	Neutropenia, pneumonia, hypertension, leukopenia, and fatigue, BNT166-related adverse events (cytokine release syndrome, bronchospasm, pyrexia)	Preliminary results - 7/20 PR and 10/20 SD by RECIST v1.1 criteria
V940 + pembrolizumab (NCT06623422)	III	Resectable stage II to IIIB (N2) NSCLC not achieving pCR after receiving neoadjuvant pembrolizumab with platinum-based doublet chemotherapy	Information not published	Pending
V940 + pembrolizumab (NCT06077760)	III	Completed resected stage II, IIIA, IIIB (with nodal involvement, N2) NSCLC.	Information not published	Pending

SD: Stable disease.

**Table 6 vaccines-13-00185-t006:** SCLC vaccine clinical trials.

Name (Clinical Trial)	Phase	Stage	Side Effects	Results
Monoclonal antibody BEC2 and BCG (NCT00003279, NCT00037713, NCT00006352 –Silva study)	III	Survival patients with limited disease SCLC	Skin ulcerations, and mild flu-like symptoms	There was no improvement in median OS (16.4 and 14.3 months *p* = 0.28), progression-free survival, or quality of life.
Pentavalent vaccine composed of KLH-conjugates of GD2L, GD3L, globo H, fucosyl GM1, and N-propionylated polysialic acid (NCT01349647)	I	Not specified	Information not published	Pending
Antigen-presenting cells pulsed in vitro with synthetic peptide corresponding to the tumor’s p53 or RAS mutation (NCT00019084 )	II	Patients with advanced cancer	Information not published	Pending
Ad.p53-DC in combination with nivolumab and ipilimumab (NCT03406715)	II	SCLC	Information not published	Pending
Dendritic cell-adenovirus p53 vaccine (NCT00049218—INGN-225 Trial)	I/II	Extensive-stage SCLC	Injections site erythema, fatigue, arthralgia, and myalgias	Primary endpoints - Safety and immune response - Clinical response - Correlation with immune response
Dendritic cell-adenovirus p53 vaccine in combination with chemotherapy with or without all trans retinoic acid (NCT00617409)	II	Extensive-stage SCLC	Information not published	Pending

## Data Availability

No new data were created or analyzed in this study. Data sharing is not applicable to this article.

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
