# Peer review of "Current Development of Therapeutic Vaccines in Lung Cancer"

_vaccines, 2025, doi:10.3390/vaccines13020185_

Round 1

Reviewer 1 Report

Comments and Suggestions for Authors

After careful review, I found that the manuscript provides an extensive overview of therapeutic vaccines for lung cancer, covering different approaches such as peptide/protein, DNA, dendritic cell, vector, mRNA, and allogeneic. However, while the manuscript is commendable for its breadth of information, several issues require improvement to enhance its scientific rigor and clarity.

Major Comments:

1.     The authors are suggested to analyze the reasons for clinical trial successes or failures, focusing on factors like patient selection, trial design, tumor microenvironment, and immune response variability. For example, the lack of insights into the MAGRIT and START trials limits the learning potential.

2.     The authors should add a table summarizing vaccine platform characteristics, including mechanisms, outcomes, side effects, target populations, and specific limitations. This would improve clarity and comparability.

3.     The authors are suggested to expand on challenges such as tumor heterogeneity, immune evasion, and low immunogenicity, providing specific examples (e.g., how immune checkpoint inhibitors address immune evasion) and proposing solutions.

4.     The authors are instructed to explore the synergistic effects of vaccines with existing therapies. Including examples of combination treatments would provide a more comprehensive perspective.

5.     The authors are recommended to enhance figures with detailed labels and annotations, particularly for mechanisms like antigen presentation and immune cell activation. Tables should consistently include endpoints and reasons for trial termination.

6.     The authors are suggested to include a section on emerging technologies, such as AI for neoantigen prediction and advanced delivery systems, to highlight future directions and innovation.

7.     The authors are instructed to revise the conclusion to better synthesize findings and provide actionable directions, such as improving patient selection or leveraging combination therapies.

8.     The authors should provide a more detailed discussion on SCLC to balance the manuscript and highlight its importance in therapeutic vaccine research.

Minor Comments:

1.     The authors are suggested to improve the abstract by briefly mentioning challenges and solutions to contextualize the manuscript.

2.     The authors are instructed to define abbreviations like "PR" and "PFS" when first used to avoid confusion.

3.     The authors are suggested to include recent breakthroughs, such as novel adjuvant systems and personalized neoantigen vaccines, to reflect current trends.

Author Response

Major comments

Comment 1: The authors are suggested to analyze the reasons for clinical trial successes or failures, focusing on factors like patient selection, trial design, tumor microenvironment, and immune response variability. For example, the lack of insights into the MAGRIT and START trials limits the learning potential.

I would like to formally express my agreement with the aforementioned point. The reasons behind the failure of certain clinical trials are thoroughly discussed in the expert opinion section, which spans pages 15 to 16. Additionally, we have included a conclusion and have expanded upon the expert opinion to provide greater clarity and insight.

Regarding trial design, it is observed that a substantial number of clinical trials conclude at phase 2 if they do not meet their primary endpoints. However, some trials have progressed to phase 3, including the MAGRIT study, the START trial, the START2 trial, and the INSPIRE trial, which evaluate peptide/protein vaccines. Other trials include the STOP trial for allogenic vaccines, as well as NCT06623422 and NCT06077760 for mRNA vaccines, along with the Silva study focused on small cell lung cancer. The MAGRIT study was terminated after an assessment by the Independent Data Monitoring Committee. Their evaluation indicated that the study product did not show sufficient effectiveness, as noted on page 3, paragraph 2. However, they did not provide further details regarding this conclusion. I will ensure that Table 1 is updated accordingly by changing column 5, row 4 from “results pending” to “the study was terminated.”

In the case of the START trial, it was noted that the primary endpoint revealed no change in overall survival (OS) OS 25.6 months (95% CI 22.5-29.2) in the tecemotide group versus 22.3 months (95% 19.6-25.5) in the placebo group. Similarly, both the START2 and INSPIRE trials were discontinued by the sponsor due to their failure to achieve specified primary and secondary endpoints, as stated on page 4, paragraph 1. Furthermore, the STOP trial did not meet its primary endpoint in median OS, 20.3 (95% CI, 16.8-23.7) for vaccinated group and 17.8 months (95% CI, 13.7-22.0).

We have enhanced our understanding of the tumor microenvironment (TME). For example, the study referenced in NCT01574222 demonstrated that the AdCCL21-DC vaccine was safe, with manageable adverse events. The vaccine elicited systemic immune responses in 6 out of 16 patients, targeting tumor-associated antigens (TAA), as revealed by ELISPOT assays. Additionally, there was an average increase of 3.4 times in CD8 T-cell infiltration per mm². Notably, 25% of the patients (4 out of 16) achieved stable disease by day 56. I will ensure that this valuable information is added to page 7, paragraph 1. The clinical trial NCT03546361 is currently evaluating the safety and efficacy of AdCCL21-DC in combination with pembrolizumab in patients with stage IV NSCLC, who are either EGFR/ALK wild-type after progression on a PD-L1 inhibitor or EGFR/ALK mutant after progression on TKI therapy. Preclinical studies have shown that intratumoral injection of CCL21-DC promotes autologous dendritic cell (DC) and T cell infiltration into the tumor, generating systemic immune responses against multiple tumor antigens. We will change information on table number 3, column 5, row 4 from “Intratumoral injection of CCL21-DC promotes …” to “results pending”, and I will also add patient criteria (patients after progression on a PD-L1 inhibitor or TKI therapy) to page 7, paragraph 1. We also provided our analysis regarding the influence of the tumor microenvironment (TME) on the efficacy of tumor vaccines. This information will be supported by expert opinions.

The variability in immune responses plays a crucial role in the effectiveness of vaccines. For instance, the CIMAvax-EGF vaccine has demonstrated benefits for patients with elevated serum EGF concentrations, as mentioned on page 2, paragraph 3. We discovered some discrepancies in the clinical trial (NCT02460367). It did not achieve its primary objective of inducing tumor regression, even among patients who demonstrated an immune response to the vaccine. We believe this is likely due to the vaccine's efficacy rather than the immune response itself.

We have added more information about SCLC and the DC p53 vaccine. This can be found on page 12, paragraph 1, and page 14, paragraph 1.

Comment 2: The authors should add a table summarizing vaccine platform characteristics, including mechanisms, outcomes, side effects, target populations, and specific limitations. This would improve clarity and comparability.

Thank you for your suggestions. We have improved the vaccine tables and images illustrating the mechanisms of vaccines. Additionally, we have included relevant information about the primary endpoints. For each type of vaccine, we have added specific advantages and limitations regarding their use.

Comment 3: The authors are suggested to expand on challenges such as tumor heterogeneity, immune evasion, and low immunogenicity, providing specific examples (e.g., how immune checkpoint inhibitors address immune evasion) and proposing solutions.

I support this viewpoint, and we have emphasized that tumor heterogeneity plays a crucial role, as evidenced by the clinical trial NCT02955290 that evaluated CIMAvax-EGF in combination with nivolumab. In this trial, patients with EGFR/ALK/KRAS wild-type profiles exhibited significantly better outcomes, achieving a median overall survival of 31.7 months (90% CI 5.9, NR) and a 3-year overall survival rate of 50% (90% CI 24,71). Additionally, this research also indicates that patients with PD-L1 expression levels of 1% or higher had improved outcomes, with a 3-year OS rate of 38% (90% CI 12, 63) and a 3-year progression-free survival (PFS) rate of 38% (90% CI 12, 63). This information has been incorporated into the manuscript and can be found on pages 2 and 3 under the subtopic Epidermal Growth Factor.

Comment 4: The authors are instructed to explore the synergistic effects of vaccines with existing therapies. Including examples of combination treatments would provide a more comprehensive perspective.

We have expanded our understanding of combination treatments involving tumor vaccines for non-small cell lung cancer (NSCLC), as outlined in recent clinical trials. Notably, we have included details on CIMAvax-EGF used in conjunction with nivolumab, as well as tecemotide combined with bevacizumab for maintenance therapy related to peptide vaccines. This information is presented in Table 1.

Additionally, we highlight the clinical trial NCT03353675, which examines first-line chemotherapy in combination with TG4010 and nivolumab. Another clinical trial evaluated TG4010 in association with chemotherapy, but it failed its primary endpoint.  The clinical trial, NCT02879760, assesses the MG1-MAGEA3 with the AdMAGEA3 vaccine alongside with pembrolizumab. Furthermore, clinical trial NCT02285816 evaluates the efficacy of MG1 Maraba/MAGE-A3, both with and without adenovirus vaccines incorporating transgenic MAGE-A3 insertion for vector-based treatments. This information is presented in Table 2. It is important to note that the results of these last two trials are still pending.

The clinical trial NCT03546361 is currently investigating AdCCL21-DC combined with pembrolizumab for dendritic cell vaccines. This information is included in Table 3.

Moreover, clinical trial NCT00601796 involves GM.CD40L bystander cells mixed with an equivalent number of the two allogeneic tumor cell lines, NCI-H1944 and NCI-H2122, in combination with cyclophosphamide and tretinoin for allogeneic vaccines. This information is included in Table 4.

Trials NCT06623422 and NCT06077760 are evaluating V940 mRNA vaccines alongside pembrolizumab; however, the results of these trials are also still awaited. This information is found in Table 5.

Regarding small cell lung cancer (SCLC), we have previously mentioned Monoclonal antibody BEC2 and the BCG vaccine, as well as Ad.p53-DC in conjunction with nivolumab and ipilimumab. Additionally, the dendritic cell-adenovirus p53 vaccine is being evaluated in combination with chemotherapy, with or without all-trans retinoic acid. Please note that the results of these studies are also pending. This information can be found in Table 6.

Comment 5: The authors are recommended to enhance figures with detailed labels and annotations, particularly for mechanisms like antigen presentation and immune cell activation. Tables should consistently include endpoints and reasons for trial termination.

Thank you for your recommendation. As mentioned earlier in comment 2, we have improved the vaccine tables and added images that illustrate the mechanisms of how vaccines work. Additionally, we have included information about the main endpoints. For each type of vaccine, we have highlighted the specific benefits and drawbacks associated with their use. You can find detailed information in the manuscript related to each vaccine.

Comment 6: The authors are suggested to include a section on emerging technologies, such as AI for neoantigen prediction and advanced delivery systems, to highlight future directions and innovation.

Thank you for your suggestions. We have included a specific topic on neoantigen vaccines, which can be found on page 11. We have briefly discussed AI in neoantigen prediction, and additional information is available in expert opinions.

Comment 7: The authors are instructed to revise the conclusion to better synthesize findings and provide actionable directions, such as improving patient selection or leveraging combination therapies.

Thank you for your guidance. We have revised the expert opinion and conclusion and provided actionable recommendations for implementing tumor vaccines. These include enrolling patients in the early stages of the disease, combining the vaccines with immune checkpoint inhibitors, testing the immune response to determine whether to continue the vaccine, and identifying better tumor markers to predict who will benefit from the vaccines. Additional information can be found in the expert opinion.

Comment 8: The authors should provide a more detailed discussion on SCLC to balance the manuscript and highlight its importance in therapeutic vaccine research.

Thank you very much for your recommendation. We have expanded our discussion on SCLC. This information can be found on page 12, paragraph 1.

Minor comments

Comment 1: The authors are suggested to improve the abstract by briefly mentioning challenges and solutions to contextualize the manuscript.

Thank you for your recommendation. We have revised the abstract to better emphasize future directions.

Comment 2: The authors are instructed to define abbreviations like "PR" and "PFS" when first used to avoid confusion.

Thank you for your guidance. We have defined the abbreviations and eliminated any repetitions.

Comment 3: The authors are suggested to include recent breakthroughs, such as novel adjuvant systems and personalized neoantigen vaccines, to reflect current trends.

Thank you for your recommendation. We have added this information to the expert opinion.

Reviewer 2 Report

Comments and Suggestions for Authors

In the present review the authors address the topic of cancer vaccines in the therapy of lung cancer.

Even if the description of the current trials is rather comprehensive, in my opinion a more detailed introduction and discussion is needed to improve and complete the manuscript.

Here I summarize my observations and requests:

The introduction is too short: the goal of a review is helping the reader to shortly know all the elements need to generally understand the argument.

A general introduction regarding the two types of cancer that the authors include in the work is lacking (general information about characteristics of the diseases, current more eligible therapies, recurrence of metastatic disease…)

In this context authors can also cite the for example the most appropriate therapy in presence of metastases (i.e nanovaccines approaches PMID 39609851).

Moreover, a summary paragraph in which they discuss how the different stages of the disease  and the microenvironment impact on the efficacy and on the applications of the different types of vaccines  would be appreciated.

It could be interesting underline if there are some predictive serological and/or immunological factors that can direct the therapy toward one or another type of vaccines: this is important in the scope of the personalized medicine.

In the introduction the authors need to add more recent references.

In the paragraph MessengerRNA they can add the review PMID 39798545.

Probably merges the figures representing the mechanisms into one (figure /table) in which the authors can also resume pro and cons of each vaccine can help the readers.

The authors can cite also embrionic stem cell-based vaccines (see Sci Rep 2024 doi: 10.1038/s41598-024-83932-0). 

A conclusive paragraph on the trials presented and a glance of next steps needed to face the still unreached medical needs in the context will be appreciated.   

Author Response

In the present review the authors address the topic of cancer vaccines in the therapy of lung cancer.

Even if the description of the current trials is rather comprehensive, in my opinion a more detailed introduction and discussion is needed to improve and complete the manuscript.

Here I summarize my observations and requests:

The introduction is too short: the goal of a review is helping the reader to shortly know all the elements need to generally understand the argument.

A general introduction regarding the two types of cancer that the authors include in the work is lacking (general information about characteristics of the diseases, current more eligible therapies, recurrence of metastatic disease)

Thank you for your recommendations. We have enhanced the introduction and discussion sections. Furthermore, we have incorporated additional information regarding two types of cancer, as well as insights into therapies and future directions.

In this context authors can also cite the for example the most appropriate therapy in presence of metastases (i.e nanovaccines approaches PMID 39609851).

We appreciate your valuable suggestions. We have briefly included the article "Personalized nanovaccines for treating solid cancer metastases" in our review. For further insights, we recommend consulting expert opinions on this topic.

Moreover, a summary paragraph in which they discuss how the different stages of the disease and the microenvironment impact on the efficacy and on the applications of the different types of vaccines would be appreciated.

We have enhanced our understanding of the tumor microenvironment. For example, the study referenced in NCT01574222 demonstrated that the AdCCL21-DC vaccine was safe, with manageable adverse events. The vaccine elicited systemic immune responses in 6 out of 16 patients, targeting tumor-associated antigens (TAA), as revealed by ELISPOT assays. Additionally, there was an average increase of 3.4 times in CD8 T-cell infiltration per mm². Notably, 25% of the patients (4 out of 16) achieved stable disease by day 56. I will ensure that this valuable information is added to page 7, paragraph 1. The clinical trial NCT03546361 is currently evaluating the safety and efficacy of AdCCL21-DC in combination with pembrolizumab in patients with stage IV NSCLC, who are either EGFR/ALK wild-type after progression on a PD-L1 inhibitor or EGFR/ALK mutant after progression on TKI therapy. Preclinical studies have shown that intratumoral injection of CCL21-DC promotes autologous dendritic cell (DC) and T cell infiltration into the tumor, generating systemic immune responses against multiple tumor antigens. I will change information on table number 3, column 5, row 4 from “Intratumoral injection of CCL21-DC promotes …” to “results pending”, and I will also add patient criteria (patients after progression on a PD-L1 inhibitor or TKI therapy) to page 7, paragraph 1.  We also provided our analysis regarding the influence of the tumor microenvironment (TME) on the efficacy of tumor vaccines. This information will be supported by expert opinions.

It could be interesting underline if there are some predictive serological and/or immunological factors that can direct the therapy toward one or another type of vaccines: this is important in the scope of the personalized medicine.

Thank you so much for your suggestions we have added specific information about immunological factors such as expression PD-L1 can be a marker for vaccine efficacy as demonstrated in the clinical trial NCT02955290 that evaluated CIMAvax-EGF in combination with nivolumab. The information has been incorporated into the manuscript and can be found on pages 2 and 3 under the subtopic Epidermal Growth Factor

We also acknowledge that variations in immune responses play a crucial role in the effectiveness of vaccines. For instance, the CIMAvax-EGF vaccine has demonstrated beneficial effects in patients with elevated serum EGF levels, as mentioned on page 2, paragraph 3. We discovered some discrepancies in the clinical trial (NCT02460367). It did not achieve its primary objective of inducing tumor regression, even among patients who demonstrated an immune response to the vaccine. We believe this is likely due to the vaccine's efficacy rather than the immune response itself. This information is also highlighted in the text on page 8, paragraph 1.

We have outlined the distinct advantages and disadvantages linked to each type of vaccine. Detailed information is available in the corresponding manuscript for each vaccine.

In the introduction the authors need to add more recent references.

Thank you for your recommendations. We have included recent references in the manuscript regarding treatments for NSCLC and SCLC. This information can be found on pages 1 and 2 for NSCLC and on page 12 for SCLC.

In the paragraph mRNA they can add the review PMID 39798545.

Thank you for your recommendation. We have included the manuscript "Clinical Advances of mRNA Vaccines for Cancer Immunotherapy" in our review. Further information can be found on page 10, paragraph 1.

Probably merges the figures representing the mechanisms into one (figure /table) in which the authors can also resume pro and cons of each vaccine can help the readers.

Thank you for your suggestions. We have improved the vaccine tables and images illustrating the mechanisms of vaccines. Additionally, we have included relevant information about the primary endpoints. For each type of vaccine, we have highlighted the specific benefits and drawbacks associated with their use. You can find detailed information in the manuscript related to each vaccine.

The authors can cite also embryonic stem cell-based vaccines (see Sci Rep 2024 doi: 10.1038/s41598-024-83932-0). 

Thank you for your valuable recommendations. We have included a brief reference to the manuscript titled “The Efficacy of an Embryonic Stem Cell-Based Vaccine for Lung Cancer Prevention Depends on the Undifferentiated State of the Stem Cells” in our review. Further information is available in Expert Opinion.

A conclusive paragraph on the trials presented and a glance of next steps needed to face the still unreached medical needs in the context will be appreciated.   

Thank you for your guidance. We have revised the expert opinion and conclusion and provided actionable recommendations for implementing tumor vaccines. These include enrolling patients in the early stages of the disease, combining the vaccines with immune checkpoint inhibitors, testing the immune response to determine whether to continue the vaccine, and identifying better tumor markers to predict who will benefit from the vaccines. Additional information can be found in the expert opinion.

Reviewer 3 Report

Comments and Suggestions for Authors

This is an excellently written, presented and updated review on the current development of therapeutic vaccines in lung cancer to improve outcomes for patients diagnosed with lung cancer.

            It includes a number of well designed and presented informative tables and figures.

            The paper ends with an Expert opinion as  a summary and reflection on the present status of the development of those vaccines, which is challenging and of rather limited success so far, 

Author Response

This is an excellently written, presented and updated review on the current development of therapeutic vaccines in lung cancer to improve outcomes for patients diagnosed with lung cancer.

 It includes a number of well designed and presented informative tables and figures.

 The paper ends with an Expert opinion as  a summary and reflection on the present status of the development of those vaccines, which is challenging and of rather limited success so far, 

Thank you very much for your review.

Round 2

Reviewer 1 Report

Comments and Suggestions for Authors

The revised manuscript addresses all the concerns raised in the previous submission. The current version is suitable for publication in Vaccines.

Reviewer 2 Report

Comments and Suggestions for Authors

The authors reply to all my requests so improving the quality and readability of the manuscript